# Fast Mass-Production of Medical Safety Shields under COVID-19 Quarantine: Optimizing the Use of University Fabrication Facilities and Volunteer Labor

**DOI:** 10.3390/ijerph17103418

**Published:** 2020-05-14

**Authors:** Vladimir Kalyaev, Alexey I. Salimon, Alexander M. Korsunsky, Alexey A. Denisov

**Affiliations:** 1FabLab, Skolkovo Institute of Science and Technology, Moscow 121205, Russia; v.kalyaev@skoltech.ru; 2HSM lab, Center for Energy Science and Technology, Skolkovo Institute of Science and Technology, Moscow 121205, Russia; a.korsunsky@skoltech.ru; 3MBLEM, Department of Engineering Science, University of Oxford, Parks Rd, Oxford OX1 3PJ, UK; 4Shared Facilities Office, Skolkovo Institute of Science and Technology, Moscow 121205, Russia; a.denisov@skoltech.ru

**Keywords:** personal protection equipment, COVID-19, mass production

## Abstract

COVID-19 pandemic provoked a number of restrictive measures, such as the closure or severe restriction of border transit for international trading traffic, quarantines and self-isolation. This caused a series of interrelated consequences that not only prevent or slow down the spread of disease, but also impact the medical systems’ capability to treat the patients and help their recovery. In particular, steeply growing demand for medical safety goods cannot be satisfied by regular suppliers due to the shortage of raw materials originating from other countries or remotely located national sources, under conditions of quarantined manpower. The current context inevitably brings back memories (and records!) of the situation 80 years ago, when WWII necessitated major effort directed at the rapid build-up of low-cost mass production to satisfy all aspects of war-time need. In the present short report we document a successful case of fast mass-production of light transparent medical safety face shields (thousands per day) realized in Skolkovo Institute of Science and Technology (Skoltech) at Fablab and Machine Shop Shared Facility (Skoltech FabLab). The demand for safety face shields by tens of hospitals in Moscow and other cities rapidly ramped up due to the need to protect medical staff during patient collection and transportation to hospitals, and within both the infected (“red”) and uninfected (“green”) zones. Materials selection for sterilizable transparent materials was conducted based on the analysis of merit indices, namely, minimal weight at given stiffness and minimal cost at given stiffness. Due to the need for permanent wear, design was motivated by low weight and comfortable head fixation, along with high production efficiency. The selection of minimal tooling in University fabrication workshops and the use of distributed volunteer labor are discussed.

## 1. Introduction

The extraordinary circumstances of COVID-19 pandemic have significantly changed the operation conditions and landscape in many fields of businesses and everyday life activities [1]. Practical logistic arrangements and delivery services, online B2B wholesale operations and supply chains are all facing disruptions that require realignment and readjustment. “Due to the lockdowns instituted to contain the further spread of the virus, e-commerce in goods has faced supply chain disruptions. Many firms have continued to experience supply challenges as a result of the suspension of manufacturing activity, decreased production and labor shortages. Those with warehousing facilities in impacted areas have faced difficult decisions about how or whether to keep manufacturing their products. The pandemic has therefore brought to the fore the vulnerabilities of supply chains and tested the ability of businesses to adjust swiftly.” The overall reduction in the annual volume of world trade is expected to be in between 13% and 32% in 2020, as COVID-19 pandemic exerts inexorable pressure on normal economic activity and life around the world [2].

New challenges emerged in the production, supply and trading of many goods such as personal safety means and protection wear, which are directly affecting the capabilities of hospitals to limit the spread of infection between medical personnel and, therefore, are currently in higher demand in US [3,4] and other countries [5]. The efficiency of goods like face filtering masks and transparent full face medical shields is discussed and impugned by some specialists [5]. It is clear, however, that even imperfect extra protection reduces the risk for medical personnel in hospitals, paramedical personnel and volunteers, and ultimately for the ordinary citizens. 

The most comprehensive review of face shields for infection control up to date [6] concludes these products were mainly considered and regulated as labor protection equipment against mechanical impacts and that at least in 2016 there were no standards (only recommendations) posing the norms on face/eye protection against infection [7]. Other aspects of personal protection equipment (PPE) safety against biohazard are covered by a number of regulations [8,9]. 

Although open transparent full face medical shields (Figure 1a,c, Appendix A) are undoubtedly less efficient than hermetic face masks protecting eyes and nose, the emergency demand for these is estimated at millions of pieces worldwide and needs to be answered within a matter of weeks. Stock reserves of medical shields are limited and many ready mask substitutes from sport (for divers and cyclists) or professional use (for stonemasons, woodworkers and metalworkers) are being used up, although their stock reserves are also relatively low. 

Meanwhile, stringent border controls over international trading traffic introduced by many governments in combination with obvious logistic limitations prevent fast supply from traditionally low-cost sources (China, Indonesia) with industrial infrastructure for mass-production for global demand built up over decades. The extremely disruptive nature of the current operational context conjures up memories of World War II circumstances when all available national manufacturing resources and manpower were needed to be mobilized for low-cost mass production. 

Rapid and almost total disruption of logistics took place during first 4 months (September–December 1941) of the siege of Leningrad, when the city with approximately 3 million inhabitants almost completely lost the lines of supply for food, fuel, electric power, and medical equipment, so that only the remaining of pre-war reserves and limited local resources could be used. The extraordinary efforts in restoring logistics operations slowly but gradually improved both the evacuation of civilians and the supplies for remaining city inhabitants and industry. Efficient use of local production facilities and highly limited resources resulted in extraordinary arrangements. In 1942, emergency electricity supply for the city was organized by means of five underwater copper cables (10 kV, 20.5 km each) which were laid at the bottom of Lake Ladoga to protect them from air attack. Cables were manufactured inside the sieged city, with insulation made from waxed and watermarked banknote paper remaining after the evacuation of the mint (Monetny Dvor) [10]. 

Low-cost mass fabrication practices from that epoch may turn out to be relevant, with obvious adaptation to the modern technological, communication and logistic landscape. Current pandemic situation and quarantine limitations are posing a number of challenges distinct from the circumstances of the XX century World Wars. The following aspects need to be taken into account:

Big cities which are the most affected by COVID-19, such as Milan, Madrid, London, New York, and Moscow. These are deeply deindustrialized, while external supplies of raw materials and tools are subject to delays or entirely disrupted.Human resources are mainly quarantined (self-isolated) or have minimal access to production workshops.Nevertheless, some stock reserves of raw materials remain readily available in local manufacturing plants and transit warehouses or at least in shopping centers, such as OBI, Leroy Merlin, etc. As a universal rule, big cities are also centers of academic science concentrated in universities. Materials Science and Engineering Departments support fabrication centers and laboratories equipped with traditional and modern tools for shaping metal, polymer and composite materials.CAD/CAM production paradigm suggests that a small number of designers (quarantined at home) and workers (granted access to workshops and focused on performing the most complex fabrication operations) can generate a significant volume of simple parts for further manual assembly by a community of volunteers or users on or off site. The transportation and delivery of parts and the collection of assembled ready products can be organized via automated delivery. 

Materials Science deals with fundamental or applied studies of the correlation chain “material composition–structure–performance–product design–production technology”. A logical extension of this chain under severe supply constraint is to incorporate issues of availability and production efficiency into consideration. This approach has recently become reflected in reports of hand-made face shields [11] and 3D-printed adaptors for PPE helmets [12] fabricated in university labs during COVID-19 pandemic. 

The unusual circumstances of COVID-19 crisis represent a specific situation where applied scientific evaluation must lead to rapid practical results. The solution of a practical task becomes the target of R&D effort; e.g., to organize *fast mass production* of relatively simple products under severe constraints due to limited material resources and manpower. 

We document a successful case of fast mass-production (thousands of items per day) of light transparent medical safety face shields realized in Skoltech Fablab and Machine Shop Shared Facility. These safety face shields were in peak demand by tens of hospitals in Moscow City and regional clinics to protect medical staff in both infected (“red”) and clean (“green”) zones. Ashby’s materials selection algorithm [13] was applied in respect of sterilizable transparent material to satisfy the performance indices for minimal weight at given stiffness and minimal cost at given stiffness. 

By the application of “as simple as possible” design principle to medical face shields, production was devised to use small and medium scale laser cutters widely available in university workshops and fabrication laboratories. The design was targeted to enable permanent wear of this means of protection, via the reduction of weight and providing comfortable fixation to head. Aspects of minimal tooling and the use of “assembling by final user” or “distributed volunteer labor” and sterilization are discussed. 

## 2. Requirements and Constraints

Up to date, only flexible recommendations and guidelines exist on the design and materials which are summarized [14] as following: 

“Face shields are commonly used as an infection control alternative to goggles. As opposed to goggles, a face shield can also provide protection to other facial areas. To provide better face and eye protection from splashes and sprays, a face shield should have crown and chin protection and wrap around the face to the point of the ear, which reduces the likelihood that a splash could go around the edge of the shield and reach the eyes. Disposable face shields for medical personnel made of light weight films that are attached to a surgical mask or fit loosely around the face should not be relied upon as optimal protection.”

The well-known prototype of the designed product is exemplified in Figure 1a. A full face shield protects the wearer’s eye mucous lining from direct flow of infected aerosol, while nose and mouse should benefit from the additional defense by tissue face mask, on the assumption that viral aerosol cannot penetrate the skin barrier, so that only mucosa are at risk. A readily affordable substitute, such as one of the variants of construction workers’ protection mask (Figure 1b), as well as professional medical goggles, defend both against direct flow of infected matter from the patient and the suspended aerosol most important in closed rooms of “dirty” (red) zones in hospitals. Goggles worn for many hours cause skin compression, scratching and irritation at the nasal bridge, upper cheeks and forehead, thus creating locations with increased potential risk of infection. Disposable positive pressure isolation suit provides the highest (but not absolute) protection and it was actually used in “red” zones or in heavily infected spaces like residences for elderly persons where high morbidity rate was detected. This is an expensive and relatively scarce solution and it is difficult for it to be applied everywhere, especially in the peak period. This poses a difficult choice for medical personnel and hospital management in the selection of the best type of PPE. Nevertheless, it appears that manipulations carried out with patients in open spaces during evacuation to hospital or disinfection operations can be more safely performed using full face shields. Below we discuss the design, materials selection, technology, and production efficiency of the full face shield devised in FabLab of Skoltech (Figure 1c and Appendix A).

The price of professional industrially manufactured full face shield reaches tens of USD from local suppliers in USA and UK, while lead time of several weeks for Chinese products with the price of units of USD per 1 pcs is not acceptable during a pandemic (a short analysis of current prices, designs and materials are presented in Appendix A). Relatively cheap (units of USD) and easily affordable protection masks (exemplified in Figure 1b) made from polycarbonate glass are designed for the protection against impact of metal and stone debris, making it heavier, while their optical characteristics do not satisfy the end user requirements in the medical context, since the clarity and transparency for fine hand operations are modest. The weight of commercially available products may reach 380 ± 80g. 

The total weight of the devised full face shield met the overall objective of 36 ± 3 g (0.3 mm thickness, general conditions) and 56 ± 3 g (0.5 mm thickness, enlarged for extra protection) in different versions, depending on the shield dimensions. Thus, it is much lighter than the industrial analogues available on the market. The weight of the full face shield must be accounted for in the context of head movement by personnel, since it needs to be balanced by the distributed force of the forehead strip and elastic band assembly, and ultimately transmitted through neck muscles. Head and shoulder loading from the lighter face shield is several times lower for the devised product than for its industrial analogues. 

## 3. Materials Selection

The practice of materials selection for face shield visors was thoroughly reviewed in [7] and the use of polycarbonate, propionate, acetate, polyvinyl chloride, and polyethylene terephthalate glycol (PETG) was justified from clarity (acetate), economics (PETG) and popularity (polycarbonate) points of view. 

Materials selection for a transparent medical face shield represents a relatively simple problem in the context of Ashby materials selection paradigm, as implemented in the educational CES Edu Pack 2019 software [15]. Systematic procedure requires the following steps: (i) translation of design requirements, (ii) screening against the material attribute limits, (iii) ranking of materials in terms of performance indices; and (iv) expert assessment and local testing. This was carried out as follows:

### 3.1. Translation

**Function:** Stiff panel (plate) resisting the bending force

**Objective:** (a) Minimize mass; (b) Minimize cost

**Constraints:***Non-negotiable constraints*: * transparency: transparent or optical quality * non-allergic and non-toxic in the contact with skin *Area AxB is specified * must not be brittle

***Negotiable constraints***: * must withstand limited bending force with small distortions * must not yield, buckle or failure under own weight and limited bending force

**Free variables: *** plate thickness * material choice

### 3.2. Screening 

The constraint on transparency significantly reduces the number of candidates from more than 4000 down to 197 as shown in the Figure 2a. Candidate materials passing the “transparency” filter represent different classes of materials (Figure 2b) including fibers and particulates, and technical ceramics like sapphire and quartz, and various glasses. Ceramics and glasses will be further excluded because of brittleness and obvious technological difficulties in the application of shaping processes. The limitation on material class (polymers and elastomers are passing this filter) further cuts down the number of candidates (Figure 2c)—only 130 materials are considered at the ranking stage.

### 3.3. Ranking

The performance indices relevant for minimal mass $\rho \cdot E_{f}^{-\frac{1}{3}}$ and cost $C_{m}\cdot \rho \cdot E_{f}^{-\frac{1}{3}}$ of bent panel were chosen as the axes for Ashby chart (Figure 3a). Here *ρ* is density, *E_f_* is flexural modulus and *C_m_* is the price per unit mass. Left bottom corner is the region of interest corresponding to light cheap material solutions and the set of 16 candidate materials in that region is depicted in Figure 3b. Red dashed line represents the envelope for best weight (SAN) and cost (SMMA) solutions. 

### 3.4. Expertize and Testing 

The list of the most attractive candidates is given below in Table 1 in the order of ascending value of cost per unit stiffness. Four candidates namely SMMA, PET, SAN, and PS are durable in water and weak organic solvents, acids and alkalis; readily sterilizable (except autoclave); and appropriate for use in contact with human skin, and they also seem to be the most recommended while the final choice is to be defined after the analysis of local prices, stock resources and available equipment for fast shaping. The PET of optical quality (that is an extra benefit) in ready sheets having thickness 0.5 mm and 0.3 mm for shortened “S” version has been finally chosen due to the fact that it was the most affordable for the Skoltech’s FabLab in March and April, 2020. This material scores second place in respect to cost saving, being, however, at least 30% less attractive in respect to weight than polystyrene.

Appendix A contains the evidences of using PET, PETG (glycol-modified polyethylene terephthalate), polyester (that is actually PET), and PPE (Polyphenylene Ether) to embody various designs of transparent visor. The forehead holders are very diverse both from design and materials points of view; they often have elastic bands for adjustments, though the sponges and rigid holders are also represented (the latter are made of polycarbonate, polypropylene and others). Thus, the choice of PET is both scientifically and practically rationalized. 

## 4. Production Requirements for Design Cutting and Assembling Operation Protocol

The guidelines of low-cost mass production listed below are well known but difficult to be universally applied for all products; however, it was historically proven in economics of wars that they are efficiently applicable in the situation when the product is relatively simple and commercial issues such as sales and profits are not involved. Nevertheless, high motivation, creativity, labor discipline and strong management are required to reach the desired records of fast low-cost mass-production.

The guidelines for fast, low-cost mass-production are:All designs must be as simple as possible. This means both minimal number of parts and simplest design of each part.All the materials used should be cheap and widely available for sale.All technologies applied must be as simple as possible to achieve the highest productivity relying on the simplest tooling for shaping and no tools for assembling using unqualified end-users or volunteers, such as oldsters, homemakers, and teenagers.The number of technological steps, pre- and post-treatments, must be minimized. This includes material synthesis, shaping and joining or assembling.The transportation of parts must be minimized along the production chains.

Following these guidelines, the design was based mainly on the cutting of ready sheets of PET and elastic fabric band. The layout of the pattern ready for laser cutting of the front visor and forehead strip is represented in the Figure 4. This pattern was used to cut PET sheets having 0.5 mm thickness (or 0.3 mm version is 35 mm shorter and 30 mm narrower). This material has been justified in terms of optical quality and minimal cost (although, other candidates provide advantages in terms of minimal mass) above, and it is relatively affordable in local conditions. Other materials from Table 1 are merely suitable if affordable from local suppliers since they also possess adequate rigidity. When changing the material, one needs to experimentally adjust the laser power and cutting speed only. Drawings do not need to be changed and that is an important and obvious advantage. 

Fixing the back elastic band for the design shown is 20 mm wide for the purpose. It can be changed to another width available in local conditions. The length of the band is to be adjusted for optimal personal comfort by a user.

## 5. Tools and Parts

In the 21st century, public workshops, fabrication departments, laboratories, or workshop facilities in universities, at schools, and in creativity clubs for children and youth became widely accessible. Almost all of them have CAD/CAM equipment such as 3D printers and laser cutters. Any cheap CO2 laser cutter can be used for manufacturing of the design given (face shield and forehead strap). For the elastic band, an option of cutting with a hot knife requires intensive exhaust ventilation. That is hard to implement at home in a city, but it is still a reasonable manner outside or in a countryside workshop in a warm climate with the personnel having breath protection means. 

The minimal set of equipment includes: A laser cutter (available in a huge number of workshops. It is also possible to shape the parts using both a cutting press or a water-jet cutter) of any power, though the mastered regime suggests that optimal performance is achieved at 100 W on PET;We recommend elastic band cutting using regular scissors (a soldering iron as a hot knife can be used ONLY if you can provide adequate ventilation and proper PPE)

Equipment and recommendations for the production:

1. Cutting the sheet material for the front visor and forehead strip. 

A CO2 laser cutter with any characteristics. If the maximum performance is required, it is optimal to use a nominal 100 W laser and an average speed of 105 ± 10 mm/s (depending on actual layout of cut part within the sheet area) at 70% of power to avoid laser source degradation. A SYNRAD FSTI100SFB, 48.3V/21A laser tube was used in the present case. If the laser cutter can maintain the quality of curved surfaces at a higher speed and the laser lifetime can be consumed, it is advisable to increase both characteristics up to the limit to be experimentally found.

One should expect a production rate of about six cut parts per 4 min 30 s at 900 × 600 mm at each laser cutter. One person can operate two laser cutters simultaneously, which results in up to 130 sets in an hour (fume-extraction time is added). A smaller machine will decrease the productivity.

2. Cutting of fabric elastic band (or rubber ribbon).

Hot knife for cutting synthetic fabric or fabric elastic bands to simultaneously cut and secure the cut edge from unraveling. A cheap household soldering iron 80 … 400 W ($3) with an initially thick but manually sharpened stinger was applied to reach the resulting performance of up to one cut per 2 s. The process requires appropriate fume extraction (Figure 5).

CAM hot knife that is available in specialized sewing workshops. 

Conventional scissors can be used to significantly accelerate the cutting process up to more than 30 cuts per minute while cutting three tapes simultaneously. A guillotine-type hand cutter (60 cuts per minute, 6 tapes at the same time) is also an option. Mechanical cutting of fabric elastic band, however, deteriorates the performance of fabric elastic band due to unraveling. 

3. Assembling the front visor, forehead strip and elastic band. In all types of design, this is done manually.

Die cutters or stamping presses driven by an electromagnetic, pneumatic, hydraulic or mechanical actuator working in-line and capable to cut thousands of pieces per working hour are undoubtedly much more performant. However, it requires the design and manufacture of dice, sharpening and hardening, and ongoing maintenance to maintain performance. The production and repairing of lasting cutting dies, which as a rule are subjected to thermally-induced tool wear [16], is the bottle neck stage in terms of time in the current situation. Under restrictive lockdown conditions, this route presents significant challenges, and at the very least would slow down the build-up of production. Moreover, the scrap fraction for stamping presses is so high that it is not acceptable in the situation of limited supply. Finally, the prices for laser cutters and lasting dies are comparable and reach a few thousands of USD, while stamping presses are much rarer in the university fabrication laboratories. 

In comparison, laser cutting has the advantage of being already available in the lab, along with expertise required to configure and operate it. Digital manufacturing instructions can be prepared and put into operation within a matter of an hour and optimized along the way without having to re-manufacture dice. 

## 6. Production Efficiency and Labor of Distributed Volunteers

Given the purpose of the work in the COVID-19 circumstances, it is likely that the workshop management is able to allocate 100% of the time for free. After simple training in the basics of labor safety rules and the practice of “one button " method for safe and effective operation, an operator can service either two units of laser cutters at a time (if they are installed side by side) or one laser cutter, but clean the corners of the face shield from the residuals of sheet material and protective transport film present on the surface of PET sheet meanwhile.

Formula (1) adopted from [13] covers main terms of production costs per product unit.
(1)$$C_{s}=\frac{m\cdot C_{m}}{\left(1-f\right)}+\frac{C_{t}}{n}\left\{\mathrm{Int}\left(\frac{n}{n_{t}}+0.51\right)\right\}+\frac{1}{\dot{n}}\left(\frac{C_{c}}{L\cdot t_{wo}}+{\dot{C}}_{oh}\right)$$
where the first term corresponds to direct material costs (*m*—mass of unit, *C_m_*—cost of material per unit mass), and *f* is scrap fraction.

The second term is the tooling (or dedicated tools) costs (*C_t_*—the cost of the set of wearable tools (laser source in case), *n*—the number of units in the batch, *n_t_*—the number of units “that a set of tooling can make before it has to be replaced, and ‘**Int**’ is the integer function. The term in curly brackets simply increments the tooling cost by that of one tool-set every time *n* exceeds *n_t_*”.

The third term is related to the time factor. A non-dedicated capital tool having *C_c_* (laser cutter in our case) is to be wrote-off in *t_wo_* at loading with the fraction of productive time *L* providing the production rate *ń*. The overhead rate ${\dot{C}}_{oh}$ that is difficult to precisely estimate for a university fabrication laboratory contributes when divided by production rate. On the other hand, the average cost of electric energy consumed per working day and labor costs with a reasonable multiplication factor can be applied.

In Table 2, we present the values of cost components used in the estimation of costs per unit face shield. The calculations of costs assume the use of two laser cutters being loaded and operated by a single worker. The second worker mechanically grinds and cleans the corners and edges of PET parts, and performs disinfection and packing operations.

Finally, our estimation for the production costs per unit shield in the current circumstances and Skoltech’s FabLab conditions gives the value of 38 Rubles with nominal CAPEX amortization period (equivalent to ca. 0.5 USD) or 45 Rubles (ca. 0.6 USD) with accelerated CAPEX amortization. The calculated costs are difficult to be directly translated into competitive commercial prices in the US market, but these costs still seem to be reasonably comparable with the current level of prices. On the other hand, as we discussed in the introduction, the mobilization during COVID-19 slightly resembles the conditions of war economics, and the profitability and commercial issues must be willingly shadowed when the objective is the increased personal safety for medical and paramedical staff, volunteers and public in a wide sense.

Chronometry tests show that the productivity of 1400 fully disinfected and packed sets “front visor + forehead strip + fabric elastic band after a hot knife” are realistic in 8 h shifts for two workers. This brings 750 sets per one worker per shift (more than 90 sets per hour or 1.5 sets per minute ready for delivery) when ready elastic bands are out-sourced. The work of a single worker or cutting of elastic bands during the shift are much less efficient. 

Chronometry tests show that for Scheme A, one operator having access to the workshop (equipped with one laser cutter with a 1000 × 2000 mm desk and one soldering iron) is able make in 1 h up to 70 sets of "front visor + forehead strip + fabric elastic band after a hot knife”. Meanwhile, up to 50% of cut parts must be pre-cleaned out of residuals even when the operator has some rest.

Scheme B assumes that one operator in the workshop is using two medium-sized or small-sized cutters and one soldering iron. This scheme gains up to 150 sets per hour of "front visor + forehead strip + fabric elastic band after a hot knife” sets without cleaning, disinfection and packaging. 

The assembling, even being very technically simple for unequipped hand manipulations, takes 2–3 min per set “front visor + forehead strip + fabric elastic band after a hot knife”. To increase the productivity of workers in the FabLab, assembling operation is obliged to be left to volunteers or end-users. The ratio of manufacturing time to assembly time is 1:4, which explains the end-user (5 min each day for regular reassemble procedure for full disinfection) or volunteer involvement (less than 3 min/PCS).

Then, a volunteer collects the sets in plastic bags for further assembling at home with a production efficiency dependent on the previous hand work experience and skills (we witnessed the efficiency of more than 30 assembled shields per hour). A volunteer courier can also deliver the sets to local delivery automats reducing the number of social contacts. Then, the assembling work is carried out at home, taking into account quarantine measures. Undoubtedly, the assembling by end users can be performed at a site that takes up to 4 min for the first time and less than 2 later on.

Optimal delivery and assembling production efficiency seems to be about 250 pcs (in 1 day with intense workload) or up to 750 pcs (for 1-week delivery—to reduce logistics costs and social contacts, when assembling is being performed by few jointly quarantined persons—an average family). It is worth to note that a strong motivation component is present: It is relatively easy to make a significant contribution in the form of real help. In this case, any starting skill set is suitable. The delivery of ready products to the hospitals is performed in the same manner, i.e., delivery automats and volunteer courier community. 

## 7. Further Development and Optimization

The ability to increase the production efficiency in the scheme above is limited by the performance of laser cutters, though up to 1.5× times growth due to the higher cutting speed (more power output, stronger air flow or inert gas shielding). Production efficiency per 1 h/one operator can be enlarged up to two times if laser cutters of a large area are closely installed in the same workshop space used.

Increasing the number of cuts with a hot knife is possible if a simple mechanical device pulling a number of parallel elastic bands to cut several bands with a long hot knife is utilized. Cutting a stack of bands is not possible since the baking of bands to each other takes place. The use of a CAM hot knife would be optimal, but this equipment is rather available in specialized workshops only. 

The increase of cleaning operation performance can be realized when a special U-shaped tongue is coded to be cut on one side of sheet to facilitate the removal of the protective film. This process is extremely specific to the particular laser cutter.

The assembly takes from four to six times more time than the chain of cutting, sanding and sanitizing operations, and it is recommended to be carried out by an end-user or a volunteer. Thus, a set of 500/1500 assembly kits sets is optimal for several days of work for one volunteer. Two-hundred-and-forty-liter garbage bags (or any other) can be used repeatedly for the transportation of assembled face shields.

The assembly kit includes:-The front visors and the forehead strips-Pre-cut hot knife elastic bands, or elastic band roll for scissor cutting (no hot knives at homes!)-Assembly instructions and end-user manual-Bags for the final product packaging

## 8. Variative Design

Taking into account different anatomical structures of heads (e.g., large male and children younger than 10 y.o.), we recommend two sizes (forehead strip length)—see Appendix A. Thus, adult and kid versions are supposed for full-day wearing. Up to 1 h adult size is suitable for 4+ y.o. kids. The shape can be adapted to allow wearing the shield with a spun bond medical mask made, with a respirator and other breath protection means. It is also possible to radically reduce the area of the face shield; in this case, the product can be used as protective panoramic glasses.

The extended versions (both in up and down directions) are possible and the forehead part with a layer of polymer foam to increase the protection shown in Figure 1a is the easiest development, with the price of full sterilization loss.

## 9. Decontamination and Service Life

Sterilization of reusable PPE is an important issue like for all Healthcare facilities [14], on the other hand, due to the shortage of available resources in the peak period of the pandemic, it becomes especially critical both in terms of personal safety, economics, and sustainability. In Table 3, the sterilizability of candidate materials is addressed. The choice of PET for a full face mask is additionally justified due to the excellent sterilizability, with help from ethylene oxide and gamma radiation, while other candidates having high mechanical and economical scores (marked by the grey background in the Table 3) are much less favorable. High temperature required for steam autoclave sterilization (121 °C and 132 °C are commonly applied worldwide, though higher temperatures are also used for metal surgeon instruments) excludes almost all widespread polymers. 

A significant number of unidentified and undercounted COVID-19 cases including asymptomatic causes lasting spread of the infection [17]. Special measures must be taken along the whole production-assembling-delivery-use-reuse chain. The disinfection breaks between shifts in the Fablab are organized. Workers from different shifts are not allowed to meet or communicate during shift changes. To achieve the goals, three separated zones are organized: “A: sheets precutting”—to fit laser cutters area, “B: laser cutting and forehead strips sanding”—to manufacture the kits, and “C: cleaning, disinfection and packaging”. 

Primary disinfection before packing of ready sets “front visor + forehead strip + fabric elastic band” was specially addressed. In accordance with [18], face shields might be classified as semi-critical patient-care equipment that touches either mucous membranes, non-intact skin or noncritical patient-care equipment that touches intact skin; thus, the high-level or low-level disinfection rather than sterilization is applicable. Taking into account that the devised medical safety shields are mainly designed for use in the “green” zones of hospitals and in open spaces during the inspections or evacuation to hospital, we believe that even cleaning or decontamination levels are viewed as a realistic scenario. Additionally, for proper sterilization with ethylene oxide, low-level disinfection using various detergents and enzymatic cleaners is recommended in [18] and professional literature for low-temperature cleaning. Solar disinfection of water in PETF bottles is a well-studied issue [19] and this method potentially may be tested in future, while traditional alcohol and chlorine-based substances are suggested as a first choice during the current COVID-19 supply crisis. 

Very general guidelines are suggested for the disinfection of face shields [8] as following:

“Healthcare setting-specific procedures for cleaning and disinfecting used patient care equipment should be followed for reprocessing reusable eye protection devices. Manufacturers may be consulted for their guidance and experience in disinfecting their respective products. Contaminated eye protection devices should be reprocessed in an area where other soiled equipment is handled. Eye protection should be physically cleaned and disinfected with the designated hospital disinfectant, rinsed, and allowed to air dry. Gloves should be worn when cleaning and disinfecting these devices.”

Disinfection/cleaning recipes and methods that have been successfully and repeatedly tested for the designed shields in FabLab are as follows:-Liquid A. Water solution of ethanol C_2_H_5_OH or isopropanol CH_3_CH(OH)CH_3_ 96 wt.%.-Liquid B. Water solution of chlorhexidine C_22_H_30_Cl_2_N_10_ 0.3 wt.%.

The face shields spend 10–30 min fully immersed in the mixture of Liquid A and Liquid B, having a volume ratio A: B equal 2:1 to 2.2:1.8. To remove excessive liquid, face shields were hold 4 min vertically to let the unnecessary extra liquid go out. Wiping with microfiber cloth allows to remove laser cutting contamination; it takes 8 s on each shield. At this speed, some liquid remains on the surface and each next shield sticks to the stack. Otherwise extra liquid should be added to the side of the stack A complete set of 100 face shields and 100 head strips are sealed in an individual plastic bag with a heater. The residuals of disinfectant liquids are entrapped between PET parts, providing prolonged sustained disinfection, which is especially suitable for urgent use in hospitals. The amount of liquid stored within stack varies from 20 mL to 28 mL per 100 kits. 

The design is also specifically suitable for fast sterilization when shields are re-used. Secondary disinfection is performed as follows:
Dispose contaminated elastic band, thus “disassemble the face shield completely”, decontaminate 100% of the surface by rinsing with disinfection liquids and assemble it back with a new elastic band (three spare ones can be included into the package). It takes less than 5 min (30 s to disassemble, 2 min for disinfection, 2 min for re-assembling). If the face shield was used in the “green” zone, manual washing is recommended.Order new elastic bands in a satisfactory amount per end-user facility (usually over 500 within one building) and receive additional packages. 

Due to the use of reliable components, the service life of the shield is limited mainly by careless operation. In particular, it is important to carelessly store the " front surface on the table”; this creates scratches, interfering comfortable view. The elastic band can be used for a long time without visible degradation. Careful operation should lead to a comfortable service life of about 1 working month.

## 10. Discussion

COVID-19 pandemic resulted in a dramatic disruption of global supply chains and resulted in temporary, but acute shortage of PPE in high demand for both medical staff in hospitals and personnel of public services (shopping centers, banks, delivery and social services, including volunteers), and ultimately for ordinary citizens. A spike demand increase by hundreds of thousands to millions of pieces is difficult to satisfy merely through existing industrial facilities; despite being potentially capable of mass production, the short supply of raw materials and semi-finished products in combination with manpower quarantined at home leads to critical underperformance of the existing mechanisms. 

University fabrication laboratories and various workshops frequently equipped with CAD/CAM tools (laser cutters, 3D-printers, robotic centers) with relatively low load under “normal” operation fall into a unique category of spare, configurable resources that can be mobilized quickly and efficiently, provided a number of key methods and approaches are adopted. Their productivity can be built-up speedily to convert them into fast mass-production facilities. Special organizational measures, however, need to be taken, e.g., to minimize the presence and interaction of workers within laboratories, out-sourcing simple hand operations to volunteers, etc. Additional productivity increase can be obtained if some operations are distributed between several university workshops. Activities of this kind became omnipresent and widespread during the COVID-19 lockdown period in many countries worldwide, and the first reports appeared in scientific literature [11,12]. These reports both encourage other operators and also highlight the need for optimization by means of (a) material cost reduction via appropriate materials selection, layout and design; (b) tool cost reduction due to maximal use of existing equipment; and (c) labor cost reduction via the use of volunteers. 

We present and analyze a successful case of fast mass-production (up to 5000 items per day) of light transparent medical safety face shields in Skoltech’s Fablab. The purpose of our report is to share experience and attract attention of other university fabrication laboratories and workshops to issues of material selection, variant design, tooling, productivity optimization, and sustained decontamination. 

We believe that the principal findings reported here are widely applicable and adaptable to local conditions. The following are concise key points:Die-free cutting (direct digital fabrication) is preferred for speedy build-up of production;Simplest design with 1–3 parts and manual assembly is recommended (we recommend one or two part designs without elastic bands for short-time usage only);Relatively thin sheet (0.3–0.5 mm) of PET provides satisfactory rigidity;2–3 working shifts are recommended per working day, with a separate space for each shift to mitigate the risk of cross-contamination and to allow disinfection and cleaning;Delivery and assembly can be best left to volunteers and/or end-users (20% of assembled kits and 80% of ready-to-assemble kits in one package);Sustained disinfection can be ensured by shipping products in sealed transport bags (which is difficult with assembled items).

## 11. Field Testing

Currently, Skoltech’s FabLab produces up to 5000 completed products per working day, supplying clinics in Moscow City and a few regional ones alongside with non-medical workers who have large numbers of everyday social contacts. 

Interviews with surgeons and resuscitators yielded information that such shields are not able to provide absolute safety, but can be recommended for wider application in queueing systems, including receptions in green zones in hospitals. On the other hand, even in red zones they are actually used by surgeons during tracheostomy, intubation (often applied for COVID-19 patients) and other surgery manipulations, since the fogging of theoretically safer googles does not allow to distinguishing and manipulating tissues with necessary clarity and precision. Face shields are more comfortable in use and the factor of weight is extremely important during a COVID-19 campaign when doctors are working over 8 h dressed in PPEs suits and heavy face shields lead to hard fatigue in the last hours of the shift.

## 12. Conclusions

COVID-19 pandemic dramatically challenges society to find technical solutions for fast mass production of low-cost personal safety means to protect medical personnel and ordinary citizens. In the situation when material sources are limited by the restrictions on trading and transportation and manpower is quarantined, these technical solutions must rely on the designs suggesting the simplest tools operated by a minimal number of operators. CAM technology realized as the cutting of sheet materials by means of widely available university workshops and fabrication laboratories is viewed as optimal for fast mass production of parts to be assembled by a community of volunteers or end users at site. 

## Figures and Tables

**Figure 1 ijerph-17-03418-f001:**
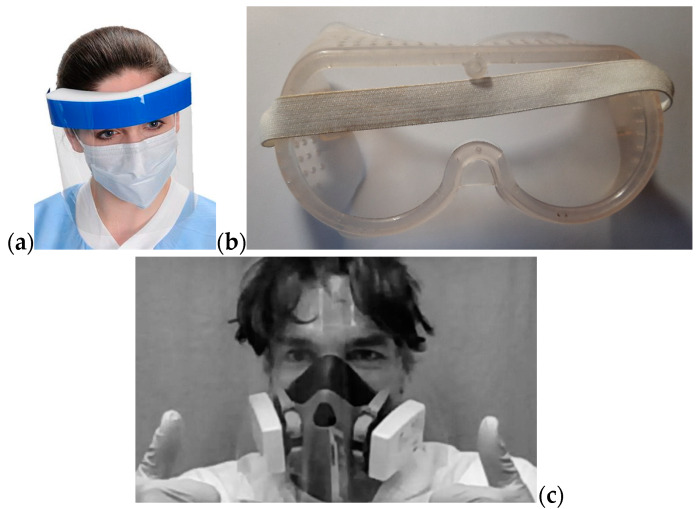
Example of personal protection equipment: (**a**) full face shield (Medical Supplies and Equipment Co., Katy, Texas 77450, USA); (**b**) construction worker goggles Archimedes 91,862 (Technoplast Ltd., St. Petersburg, Russia); (**c**) full face shield (FabLab, Skolkovo Institute of Science and Technology, Moscow, Russia)—near to invisible.

**Figure 2 ijerph-17-03418-f002:**
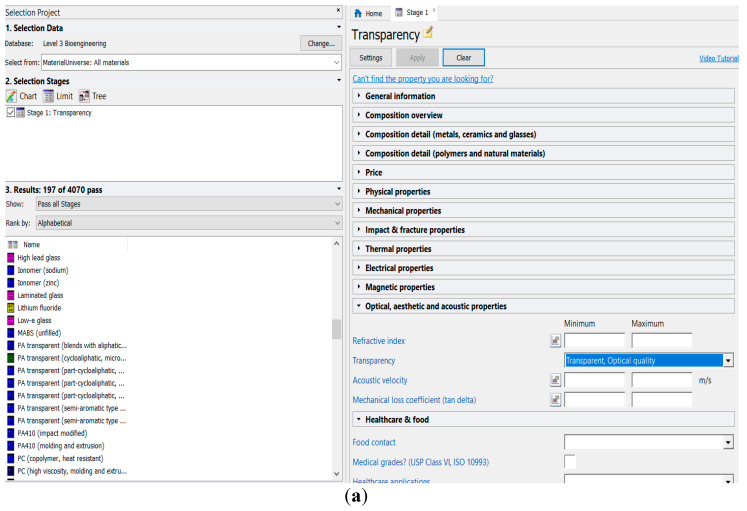
Materials selection for transparent medical face masks. Screening stage: (**a**) the list of candidate materials passed “transparency” filter; (**b**) Ashby charts for all transparent materials; (**c**) for polymers and elastomers (Charts and data from CES EduPack 2019, Granta Design Limited, Cambridge, UK, 2019 [15]).

**Figure 3 ijerph-17-03418-f003:**
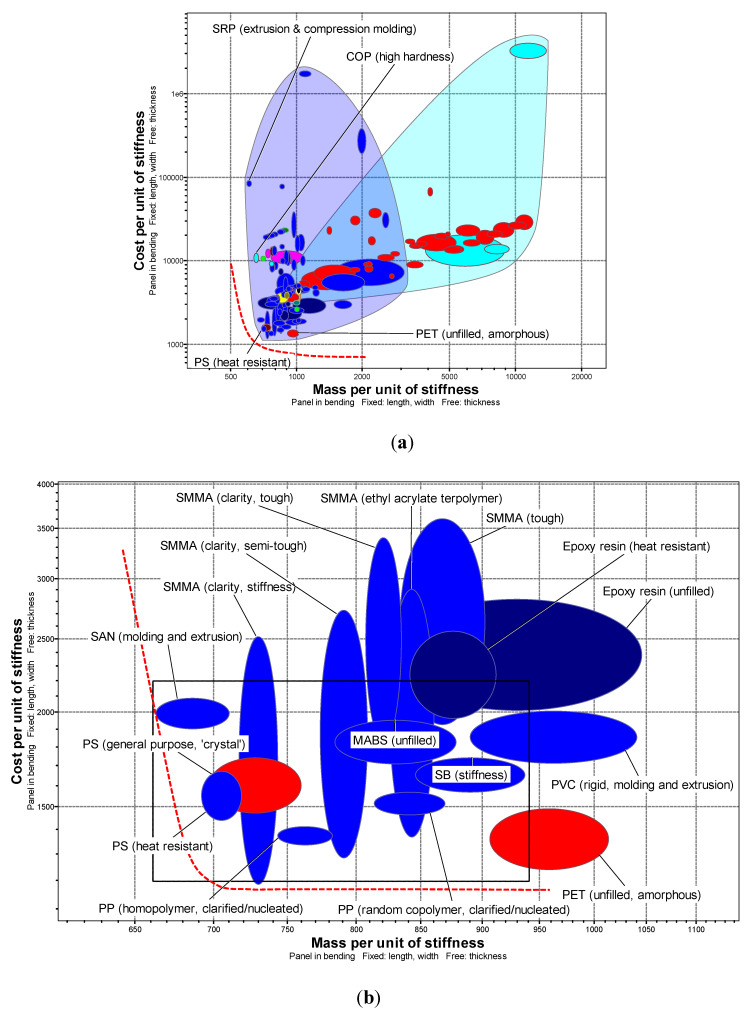
Materials selection for transparent medical face masks. Ranking stage: (**a**) 130 candidates; (**b**) 16 candidates (red—optical quality; blue—transparent). (Charts and data from CES EduPack 2019, Granta Design Limited, Cambridge, UK, 2019 [15]).

**Figure 4 ijerph-17-03418-f004:**
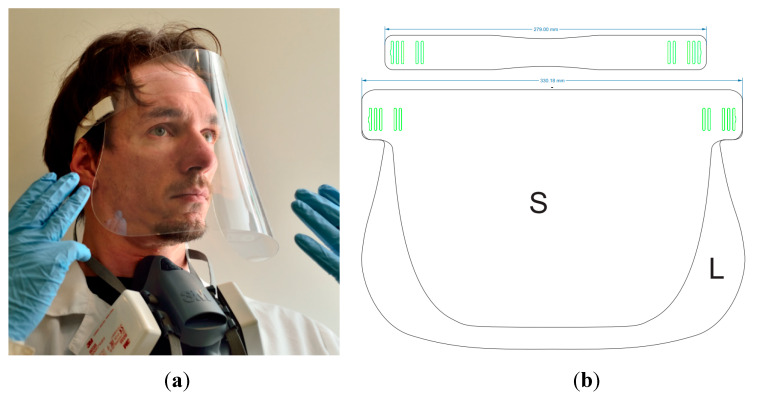
Devised full face transparent shield: stages of fabrication: (**a**) ready product; (**b**) pattern for laser cutting of L vs. S size; (**c**) laser cutting S size template example less than 8 min total; (**d**,**e**) assembling of forehead strip and front visor together with fabric elastic band; (**f**) adjusted band length to maintain comfort pressure.

**Figure 5 ijerph-17-03418-f005:**
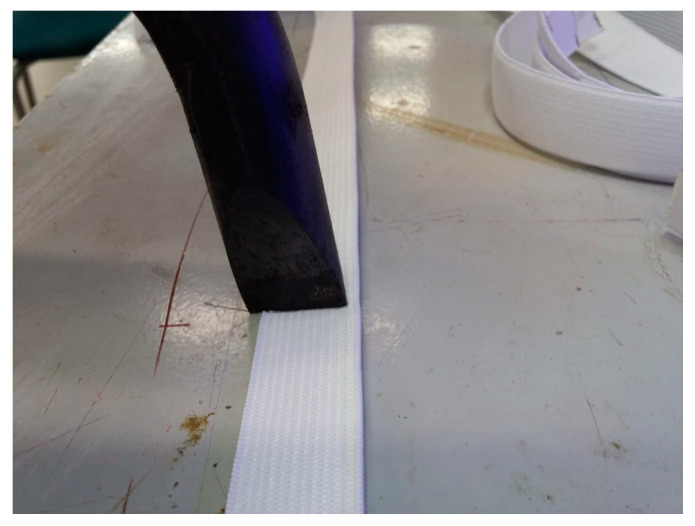
Cutting of elastic band by means of hot knife—soldering iron.

**Table 1 ijerph-17-03418-t001:** Performance of candidate materials (performance indices of two top candidates in respect of relevant performance are given in bold) (Data from CES EduPack 2019, Granta Design Limited, Cambridge, UK, 2019 [15]).

N	Name	Cost Per Unit of Stiffness (USD/(GPa^1/3^·m^3^))	Mass Per unit Stiffness (kg/(GPa^1/3^·m^3^)) (Place in the Order of Ascending Mass)	Comments
1	Styrene-methyl methacrylate copolymer **SMMA** (clarity, stiffness)	**1190–2520**	716–743 (3)	Susceptible for stress whitening
2	Polyethylene Terephthalate **PET** (unfilled, amorphous)	**1240–1490**	907–1010 (10)	Not suitable for negative temperatures
3	**SMMA** (clarity, semi-tough)	1290–2730	773–808 (5)	
4	Polypropylene **PP** (homopolymer, clarified/nucleated)	1340–1410	743–782 (4)	
5	**SMMA** (ethyl acrylate terpolymer)	1370–2910	825–859 (8)	
6	Polystyrene **PS** (heat resistant)	1440–1670	**692–718 (2)**	Poor wear and fatigue resistance
7	**PP** (random copolymer, clarified/nucleated)	1470–1570	814–869 (7)	
8	**PS** (general purpose, ‘crystal’)	147–1740	697–759 (3)	
9	Styrene-Butadiene **SB** (stiffness)	1570–1740		
10	Methyl methacrylate-acrylonitrile-butadiene-styrene **MABS** (unfilled)	1710–1950	784–878 (6)	
11	Polyvinyl chloride **PVC** (rigid, molding and extrusion)	1720–2010	890–1040 (9)	
12	Styrene acrylonitrile **SAN** (molding and extrusion)	1910–2090	**663–710 (1)**	Poor wear resistance

Note: Bold type is to highlight commonly recognized names of polymers identified in the Selection procedure above.

**Table 2 ijerph-17-03418-t002:** Inputs for the cost estimation.

N	Attribute	Value	Remarks
1	Production rate *ń*	4500 pcs/day	Three 8 h shifts (2 workers/shift)
2	Mass of ready mask *m*	60 g	
3	Scarp fraction *f*	0.15	
4	Cost of 2010 × 1250 × 0.5 mm PET sheet *C_m_*	500 Rubles	4.12 USD/kg
5	Cost of laser tube *C_t_*	2 × 48 000 Rubles	2 × 640 USD
6	Number of 8 h shifts before laser tube replacement *n_t_*	750	
7	Cost of laser cutter *C_c_*	1060 000 Rubles	2 new cutters
8	Loading fraction *L*	0.83	When fully dedicated for this particular product
9	Time to write-off	5 years—nominal2 months (allocated to project)	
10	Overhead rate ${\dot{C}}_{oh}$	Electric energy 600 Rubles per working day Elastic band, disinfection liquids, packing consumables—up to 10 Rubles per unit product	Electric power consumed 60 kWh per working day
11	Labor costs per working day	33150 Rubles including taxes and wages	3 shifts (2workers/shift)

**Table 3 ijerph-17-03418-t003:** Sterilizability of candidate materials is given in comparison with the exposure to ethylene oxide (EtO) gas and rated as excellent, good, marginal and poor. Candidates 1, 2, 6 and 12 scored highest ranks in the Section 3. Materials Selection (Data from CES EduPack 2019, Granta Design Limited, Cambridge, UK, 2019 [15]).

N	Name	Sterilizability		
		Ethylene Oxide	Gamma Radiation	Steam Autoclave
1	Styrene-methyl methacrylate copolymer **SMMA** (clarity, stiffness)	Good	Good	Poor
2	Polyethylene Terephthalate **PET** (unfilled, amorphous)	Excellent	Excellent	Poor
3	**SMMA** (clarity, semi-tough)	Marginal	Good	Poor
4	Polypropylene **PP** (homopolymer, clarified/nucleated)	Good	Poor	Good
5	**SMMA** (ethyl acrylate terpolymer)	Good	Good	Poor
6	Polystyrene **PS** (heat resistant)	Marginal	Excellent	Poor
7	**PP** (random copolymer, clarified/nucleated)	Good	Poor	Good
8	**PS** (general purpose, ‘crystal’)	Marginal	Excellent	Poor
9	Styrene-Butadiene **SB** (stiffness)	Marginal	Good	Poor
10	Methyl methacrylate-acrylonitrile-butadiene-styrene **MABS** (unfilled)	Good	Good	Poor
11	Polyvinyl chloride **PVC** (rigid, molding and extrusion)	Excellent	Marginal	Poor
12	Styrene acrylonitrile **SAN** (molding and extrusion)	Marginal	Good	Poor

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
