# Peer review of "Fast Mass-Production of Medical Safety Shields under COVID-19 Quarantine: Optimizing the Use of University Fabrication Facilities and Volunteer Labor"

_ijerph, 2020, doi:10.3390/ijerph17103418_

Round 1
Reviewer 1 Report
Dear authors,
This is a very interesting paper which has the potential to make a good contribution in the fields of medical assistance, supply chain management and health-related improvisation policies under crisis management.
I provide below some comments on how to improve the paper so as to strengthen in particular its theoretical contribution and assist you to enhance the findings.
On page 2, I would recommend that you include some contemporary references from valid media (Reuters, Economist, BBC for example) on how global supply chains have been severely disrupted by the Covid-19 pandemic. Please show some evidence with support of reference so as to avoid a, let’s say, journalistic narrative. Similarly, where have you identified information about WWII circumstances? Please cite relevant books. I totally agree with you on this historical comparison but try to be more specific. Indeed USSR’s, for example, military supply chain management has been going through massive and everyday transformations, nationally, during the siege of Leningrad and Stalingrad in terms of food, medical suppliers and clothing as several historical resources suggest. Please enrich your document with few references. At the end of page 2, I would recommend avoid presenting the arguments with numbering and (with some references) support your claims about those points. I agree with all of them but more argumentation and evidence is needed here.
Please state a central research objective for the paper using some hypothesis or some sentences. I see the practical implications what exactly do you plan to achieve and how this contributes to the existing knowledge. Please insert there at before the ‘Requirements and Constrains’ section.
I will recommend to broaden the scope of your discussion with examples from other countries/industries which are championing medical face shields. I am not an expert in the area but I assume that production experiments take place in Europe, USA and Asia within private and public institutions. On page 9, tools and parts please elaborate on the ‘cutting’ method (without laser) in terms of workforce, expertise and hygiene standards. I agree that it can be a space outside a city but can you please give a couple of examples to support you arguments? Can you also please develop the ‘Field Testing’ section. Do you consider the Fab Lab supply chain to very efficient. What type of feedback do you receive from hospitals, police and bank clerks. How can we generalize these findings for more countries which have different service encounters and climate conditions?
Please expand you conclusions on how the findings and applicability from this paper can be used in other labs, industries, countries. I am not an expert in the technical aspects of the paper but I hope these comments will help you to strengthen the technique and make it more available to other labs and user. Well done to the crew of Fablab.
To summarise: a) please provide more evidence of readings/facts in the first part from valid academic or media sources about the supply health chain crisis management b) please strengthen your discussion on your findings have applicability and can be generalize in other contexts c) please identify any limitations and expand your conclusions. There is lots of debate about the use of masks and we still know a few things about the level of protection (I am a strong supporter of universal use) so please enrich the document to make stronger argument.
All the best
Author Response
Reviewer #1. Open Review
The authors wish to thank all Reviewers for their time devoted to careful checking our manuscript, and for their helpful remarks and observations. We addressed all of them step by step and made a deep revision of the manuscript.
- On page 2, I would recommend that you include some contemporary references from valid media (Reuters, Economist, BBC for example) on how global supply chains have been severely disrupted by the Covid-19 pandemic. Please show some evidence with support of reference so as to avoid a, let’s say, journalistic narrative.
We addressed to these points in INTRODUCTION using official WTO press releases and reports as following:
The extraordinary circumstances of COVID-19 pandemic have changed significantly the operation conditions and landscape in many fields of businesses, and everyday life activities [1]. Practical logistic arrangements and delivery services, online B2B wholesale operations and supply chains are all facing disruptions that require realignment and readjustment. “Due to the lockdowns instituted to contain the further spread of the virus, e-commerce in goods has faced supply chain disruptions. Many firms have continued to experience supply challenges as a result of the suspension of manufacturing activity, decreased production and labour shortages. Those with warehousing facilities in impacted areas have faced difficult decisions about how or whether to keep manufacturing their products. The pandemic has therefore brought to the fore the vulnerabilities of supply chains and tested the ability of businesses to adjust swiftly.” The overall reduction in the annual volume of world trade is expected to lie between 13% and 32% in 2020, as COVID 19 pandemic exerts inexorable pressure on normal economic activity and life around the world [2].
- Similarly, where have you identified information about WWII circumstances? Please cite relevant books. I totally agree with you on this historical comparison but try to be more specific. Indeed USSR’s, for example, military supply chain management has been going through massive and everyday transformations, nationally, during the siege of Leningrad and Stalingrad in terms of food, medical suppliers and clothing as several historical resources suggest. Please enrich your document with few references. At the end of page 2, I would recommend avoid presenting the arguments with numbering and (with some references) support your claims about those points. I agree with all of them but more argumentation and evidence is needed here.
We addressed to these points in INTRODUCTION as following:
The extremely disruptive nature of the current operational context conjures up memories of World War II circumstances when all available national manufacturing resources and manpower needed to be mobilized for low cost mass production.
Rapid and almost total disruption of logistics took place during first four months (September – December 1941) of the siege of Leningrad, when the city with approximately 3 million inhabitants almost completely lost the lines of supply for food, fuel, electric power and medical equipment, so that only the remaining of pre-war reserves and limited local resources could be used. The extraordinary efforts in restoring logistics operations slowly but gradually improved both the evacuation of civilians and the supplies for remaining city inhabitants and industry. Efficient use of local production facilities and highly limited resources resulted in extraordinary arrangements. In 1942, emergency electricity supply for the city was organized by means of five underwater copper cables (10 kV, 20.5 km each) which were laid at the bottom of Lake Ladoga to protect them from air attack. Cables were manufactured inside the sieged city, with insulation made from waxed and watermarked banknote paper remaining after the evacuation of the mint (Monetny Dvor) [10].
- Please state a central research objective for the paper using some hypothesis or some sentences. I see the practical implications what exactly do you plan to achieve and how this contributes to the existing knowledge. Please insert there at before the ‘Requirements and Constrains’ section.
We attempted to answer this remark inserting the declaration on the objective and hypothesis in the INTRODUCTION as following
Materials Science deals with fundamental or applied studies of the correlation chain “material composition – structure – performance – product design - production technology”. A logical extension of this chain under severe supply constraint is to incorporate issues of availability and production efficiency into consideration. This approach has recently become reflected in reports of hand-made face shields [11] and 3D-printed adaptors for PPE helmets [12] fabricated in university labs during COVID-19 pandemic.
The unusual circumstances of COVID-19 crisis represent a specific situation when applied scientific evaluation must lead to rapid practical results. The solution of a practical task becomes the target of R&D effort; e.g. to organize fast mass production of relatively simple products under severe constraints due to limited material resources and manpower.
- I will recommend to broaden the scope of your discussion with examples from other countries/industries which are championing medical face shields. I am not an expert in the area but I assume that production experiments take place in Europe, USA and Asia within private and public institutions.
On page 9, tools and parts please elaborate on the ‘cutting’ method (without laser) in terms of workforce, expertise and hygiene standards. I agree that it can be a space outside a city but can you please give a couple of examples to support you arguments?
We addressed to these points in SUPPLEMENTARY MATERIALS S1 (industrial production), INTRODUCTION and DISCUSSION (production experiments in universities), TOOLS (mechanical die cutting and stamping presses). DECONTAMINATION AND SERVICE LIFE (hygiene standards) and sections.
Taking into account remarks on other production experiments we admit that efforts of similar nature taking place in Europe, Asia and USA have been well publicized through public media. However, the manner and quality of reporting is not rigorous from scientific point of view, as it lacks the detailed description and analysis that is the convention in peer-reviewed research. Nevertheless, we have been able to identify two most recent publications on related topics, relating to hand-made face shields [11] and 3D-printed adaptors for PPE helmets [12].
….
The choice of cutting method has been made based on rational arguments. It is apparent that in rapid build-up of production capacity it is important to consider optimal compromise between productivity, cost, and speed of commissioning. Given the adoption of polymer sheet already available, die punching provides optimal production mode in terms of rate and performance. However, it requires design and manufacture of dice, sharpening and hardening, and ongoing maintenance to maintain performance. Under restrictive lockdown conditions this route presents significant challenges, and in the very least would slow down the build-up of production. In comparison, laser cutting has the advantage of being already available in the lab, along with expertise required to configure and operate it. Digital manufacturing instructions can be prepared and put into operation within a matter of an hour, and optimized along the way without having to re-manufacture dice.
- Can you also please develop the ‘Field Testing’ section. Do you consider the Fab Lab supply chain to very efficient. What type of feedback do you receive from hospitals, police and bank clerks. How can we generalize these findings for more countries which have different service encounters and climate conditions?
Field testing was carried out at several hospitals in Moscow City and Region listed in Supplementary Materials. S3. Private communications of interviewed ICU practitioners were added.
- Please expand you conclusions on how the findings and applicability from this paper can be used in other labs, industries, countries. I am not an expert in the technical aspects of the paper but I hope these comments will help you to strengthen the technique and make it more available to other labs and user. Well done to the crew of Fablab.
We discuss findings and applicability in added DISCUSSION section.
DISCUSSION
COVID-19 pandemic resulted in a dramatic disruption of global supply chains and resulted in temporary, but acute shortage of PPE in high demand for both medical staff in hospitals and personnel of public services (shopping centers, banks, delivery and social services, including volunteers), and ultimately for ordinary citizens. Spike demand increase by hundreds of thousands to millions of pieces is difficult to satisfy merely through existing industrial facilities: despite being potentially capable of mass production, the short supply of raw materials and semi-finished products in combination with manpower quarantined at home leads to critical underperformance of the existing mechanisms.
University fabrication laboratories and various workshops frequently equipped with CAD/CAM tools (laser cutters, 3D-printers, robotic centers) with relatively low load under “normal” operation fall into a unique category of spare, configurable resource that can be mobilized quickly and efficiently, provided a number of key methods and approaches are adopted. Their productivity can be built-up speedily to convert them into fast mass-production facilities. Special organizational measures, however, need to be taken, e.g. to minimize the presence and interaction of workers within laboratories, out-sourcing simple hand operations to volunteers, etc. Additional productivity increase can be obtained if some operations are distributed between several university workshops. Activities of this kind became omnipresent and widespread during COVID-19 lockdown period in many countries worldwide, and first reports appeared in scientific literature [11, 12]. These reports both encourage other operators, and also highlight the need for optimization by means of a) material cost reduction via appropriate materials selection, layout and design; b) tool cost reduction due to maximal use of existing equipment; c) labour cost reduction via the use of volunteers.
- We present and analyze a successful case of fast mass-production (up to five thousand items per day) of light transparent medical safety face shields in Skoltech’s fabrication laboratory (FabLab). The purpose of our report is to share experience and attract attention of other university fabrication laboratories and workshops to issues of material selection, variant design, tooling, productivity optimization, and sustained decontamination.
- We believe that principal findings reported here are widely applicable and adaptable to local conditions. The following are concise key points:
- Die-free cutting (direct digital fabrication) is preferred for speedy build-up of production;
- Simplest design with 1-3 parts and manual assembly is recommended (we recommend 1 or 2 parts designs without elastic bands for short-time usage only);
- Relatively thin sheet (0.3-0.5 mm) of PET provide satisfactory rigidity;
- 2-3 working shifts are recommended per working day, with separate space for each shift to mitigate the risk of cross-contamination, and to allow disinfection and cleaning;
- Delivery and assembly can be best left to volunteers and/or end-users (20% of assembled kits and 80% of ready-to-assemble kits in one package) ;
- Sustained disinfection can be ensured by shipping products in sealed transport bags (which is difficult with assembled items).
- To summarise: a) please provide more evidence of readings/facts in the first part from valid academic or media sources about the supply health chain crisis management b) please strengthen your discussion on your findings have applicability and can be generalize in other contexts c) please identify any limitations and expand your conclusions. There is lots of debate about the use of masks and we still know a few things about the level of protection (I am a strong supporter of universal use) so please enrich the document to make stronger argument.
(a) Numerous public reports [1-5]
(b) See text above
(c) See text above

Reviewer 2 Report
The medical safety shield project is well presented, in a very pragmatic and clever way. It should be also very instructive for peers.
The review of commercial products should be strengthened as it may be by itself sources of inspiration and selection of materials. And if literature is available, it would be very informative to discuss average selling prices vs cost of goods of the cheapest commercial options.
The manuscript may be enriched with a cost analysis of the manufactured solution.
Even if it is not a specific / mandatory goal in the context of the Covid-19 quarantine, it should be worthwhile to know how far the selected face shield may already comply with prevailing standards.
Finally, I would change in the manuscript Sterilization into Cleaning or Decontamination since selected methods cannot garantee true sterilization (ie. reduction by 10e6 of bacteria load). And I would recommend disinfection methods which are compatible with materials and easily accessible and affordable solutions where face shields have been distributed or may be distributed.
Author Response
We appreciate the valuable remarks from Reviewer and revised our manuscript in order to address them as following:
- The review of commercial products should be strengthened as it may be by itself sources of inspiration and selection of materials. And if literature is available, it would be very informative to discuss average selling prices vs cost of goods of the cheapest commercial options.
We agree with this remark and revised our arguments in the sections REQUIREMENTS AND CONSTRAINTS
The price of professional industrially manufactured full face shield reaches tens of USD from local suppliers in USA and UK, while lead time of several weeks for Chinese products with the price of units of USD per 1 pcs is not acceptable during pandemic (a short analysis of current prices, designs and materials are presented in Supplementary materials S1). Relatively cheap (units of USD) and easily affordable protection mask (exemplified in Figure 1b) made from polycarbonate glass are designed for the protection against impact of metal and stone debris, making it heavier, while their optical characteristics do not satisfy the end user requirements in the medical context, since the clarity and transparency for fine hand operations are modest. The weight of commercially available products may reach 380 ± 80g.
and MATERIALS SELECTION adding some information to Supplementary Materials.
Supplementary Material S1 contains the evidences of using PET, PETG (glycol-modified polyethylene terephthalate), polyester (that is actually PET) and PPE (Polyphenylene Ether) to embody various designs of transparent visor. The forehead holders are very diverse both from design and materials points of view, they often have elastic bands for adjustments, though the sponges and rigid holders are also represented (the latter are made of polycarbonate, polypropylene and others). Thus, the choice of PET is both scientifically and practically rationalized.
- The manuscript may be enriched with a cost analysis of the manufactured solution.
We agree with this remark and enriched the manuscript with cost analysis in the sections PRODUCTION EFFICIENCY AND LABOR OF DISTRIBUTED VOLONTEERS
Formula (1) adopted from [13] covers main terms of production costs per product unit.
, (1)
where the first term corresponds to direct material costs (m - mass of unit, Cm – cost of material per unit mass, f - scrap fraction);
the second term is the tooling (or dedicated tools) costs (Ct – the cost of the set of wearable tools (laser source in case), n – the number of units in the batch, nt - the number of units “that a set of tooling can make before it has to be replaced, and ‘Int’ is the integer function. The term in curly brackets simply increments the tooling cost by that of one tool-set every time n exceeds nt.”;
the third term is related to the time factor. A non-dedicated capital tool having Cc (laser cutter in our case) is to be write-off in two at loading with the fraction of productive time L providing the production rate Å„. The overhead rate ÄŠoh that is difficult to precisely estimate for a university fabrication laboratory contributes when divided by production rate. On the other hand, the average cost of electric energy consumed per working day and labors costs with a reasonable multiplication factor can be applied.
In the Table 2 we present the values of cost components used in the estimation of costs per unit face shield. The calculations of costs assume the use of 2 laser cutters being loaded and operated by a single worker. Second worker mechanically grinds and cleans the corners and edges of PET parts, performs disinfection and packing operations.
Finally, our estimation for the production costs per unit shield in the current circumstances and Skoltech Fablab conditions gives the value of 38 Rubles with nominal CAPEX amortization period (equivalent to ca. 0.5 USD) or 45 Rubles (ca. 0.6 USD) with accelerated CAPEX amortization. The calculated costs are difficult to be directly translated into competitive commercial prices at US market, but these costs still seem to reasonably comparable with current level of prices. On other hand, as we discussed in the introduction the mobilization during COVID-19 slightly resembles the conditions of war economics, and the profitability and commercial issues must be willingly shadowed when the objective is the increased personal safety for medical and paramedical staff, volunteers and public in a wide sense.
Table 2
Inputs for the cost estimation
|
N |
Attribute |
Value |
Remarks |
|
1 |
Production rate Å„ |
4500 pcs/day |
Three 8h shifts (2 workers/shift) |
|
2 |
Mass of ready mask m |
60 g |
|
|
3 |
Scarp fraction f |
0.15 |
|
|
4 |
Cost of 2010Ñ…1250Ñ…0.5 mm PET sheet Cm |
500 Rubles |
4.12 USD/kg |
|
5 |
Cost of laser tube Ct |
2 x 48 000 Rubles |
2 x 640 USD |
|
6 |
Number of 8 h shifts before laser tube replacement nt |
750 |
|
|
7 |
Cost of laser cutter Cc |
1060 000 Rubles |
2 new cutters |
|
8 |
Loading fraction L |
0.83 |
When fully dedicated for this particular product |
|
9 |
Time to write-off |
5 years - nominal 2 months (allocated to project) |
|
|
10 |
Overhead rate ÄŠoh |
Electric energy 600 Rubles per working day Elastic band, disinfection liquids, packing consumables – up to 10 Rubles per unit product |
Electric power consumed 60 kWh per working day |
|
11 |
Labour costs per working day |
33150 Rubles including taxes and wages |
3 shifts (2workers/shift) |
Chronometry tests show that the productivity of 1400 fully disinfected and packed sets “front visor + forehead strip + fabric elastic band after a hot knife” are realistic at 8 h shift for 2 workers. This brings 750 sets per one worker per shift (more than 90 sets per hour or 1.5 sets per minute ready for delivery) when ready elastic bands are out-sourced. The work of a single worker or cutting of elastic bands during the shift are much less efficient.
- Even if it is not a specific / mandatory goal in the context of the Covid-19 quarantine, it should be worthwhile to know how far the selected face shield may already comply with prevailing standards.
We addressed to this valuable remark and we gave a short review of the existing US standards in the INTRODUCTION
The most comprehensive review of face shields for infection control up to date [6] concludes these products were mainly considered and regulated as labor protection equipment against mechanical impacts and that at least in 2016 there were no standards (only recommendations) posing the norms on face/eye protection against infection [7]. Other aspects of personal protection equipment (PPE) safety against biohazard are covered by a number of regulations [8, 9].
and REQUIREMENTS AND CONSTRAINTS.
Up to date only flexible recommendations and guidelines exist on the design and materials which are summarized [14] as following:
“Face shields are commonly used as an infection control alternative to goggles. As opposed to goggles, a face shield can also provide protection to other facial areas. To provide better face and eye protection from splashes and sprays, a face shield should have crown and chin protection and wrap around the face to the point of the ear, which reduces the likelihood that a splash could go around the edge of the shield and reach the eyes. Disposable face shields for medical personnel made of light weight films that are attached to a surgical mask or fit loosely around the face should not be relied upon as optimal protection.”
We emphasize that up to date full face shields are not obliged to comply to the strict norms requiring minimal or nominal mechanical and physical properties and biohazard safety, since only recommendations and guidelines exist. We state that the designed face shield seems to satisfy existing flexible regulations.
- I would change in the manuscript Sterilization into Cleaning or Decontamination since selected methods cannot garantee true sterilization (ie. reduction by 10e6 of bacteria load). And I would recommend disinfection methods which are compatible with materials and easily accessible and affordable solutions where face shields have been distributed or may be distributed.
We agree with this remark and updated our discussion and arguments in the sections DECONTAMINATION AND SERVICE LIFE.
Sterilization of reusable PPE is an important issue like for all Healthcare facilities [14], on the other hand, due to the shortage of available resources in the peak period of pandemic it becomes especially critical both in terms of personal safety, and economics, and sustainability. In the Table 3 the sterilizability of candidate materials is addressed. The choice of PET for full face mask is additionally justified due to the excellent sterilizability with help of ethylene oxide and gamma radiation, while other candidates having high mechanical and economical score (marked by the grey background in the Table 3) are much less favorable. High temperature required for steam autoclave sterilization (121°C and 132°C are commonly applied worldwide, though higher temperatures are also used for metal surgeon instruments) excludes almost all widespread polymers.
Table 3
Sterilizability of candidate materials is given in comparison with the exposure to ethylene oxide (EtO) gas and rated as excellent, good, marginal and poor are. (Data from CES EduPack 2019, Granta Design Limited, Cambridge, UK, 2019 [15])
|
N |
Name |
Sterilizability |
||
|
Ethylene oxide |
Gamma radiation |
Steam autoclave |
||
|
1 |
Styrene-methyl methacrylate copolymer SMMA (clarity, stiffness) |
Good |
Good |
Poor |
|
2 |
Polyethylene Terephthalate PET (unfilled, amorphous) |
Excellent |
Excellent |
Poor |
|
3 |
SMMA (clarity, semi-tough) |
Marginal |
Good |
Poor |
|
4 |
Polypropylene PP (homopolymer, clarified/nucleated) |
Good |
Poor |
Good |
|
5 |
SMMA (ethyl acrylate terpolymer) |
Good |
Good |
Poor |
|
6 |
Polystyrene PS (heat resistant) |
Marginal |
Excellent |
Poor |
|
7 |
PP (random copolymer, clarified/nucleated) |
Good |
Poor |
Good |
|
8 |
PS (general purpose, 'crystal') |
Marginal |
Excellent |
Poor |
|
9 |
Styrene-Butadiene SB (stiffness) |
Marginal |
Good |
Poor |
|
10 |
Methyl methacrylate-acrylonitrile-butadiene-styrene MABS (unfilled) |
Good |
Good |
Poor |
|
11 |
Polyvinyl chloride PVC (rigid, molding and extrusion) |
Excellent |
Marginal |
Poor |
|
12 |
Styrene acrylonitrile SAN (molding and extrusion) |
Marginal |
Good |
Poor |
Significant number of unidentified and undercounted COVID-19 cases including asymptomatic causes lasting spread of the infection [17]. Special measures must be taken along whole production-assembling-delivery-use-reuse chain. The disinfection breaks between shifts in the Fablab are organized. Workers from different shifts are not allowed to meet or communicate during shift changes. To achieve the goals 3 separated zones are organized: “A: sheets precutting” – to fit laser cutters area, “B: laser cutting and forehead strips sanding” – to manufacture the kits, and “C: cleaning, disinfection and packaging”.
Primary disinfection before packing of ready sets “front visor + forehead strip + fabric elastic band” was specially addressed. In accordance with [18] face shields might be classified as semi-critical patient-care equipment that touches either mucous membranes or non-intact skin or noncritical patient-care equipment that touches intact skin, thus the high-level or low-level disinfection rather than sterilization is applicable. Taking into account that the devised medical safety shields are mainly designed for the use in the “green” zones of hospitals and in open spaces during the inspections or evacuation to hospital we believe that even cleaning or decontamination levels are viewed as the realistic scenario. Additionally, to the proper sterilization with ethylene oxide, low-level disinfection using various detergents and enzymatic cleaners are recommended in [18] and professional literature for low temperature cleaning. Solar disinfection of water in PETF bottles is a well-studied issue [19] and this method potentially may be tested in future, while traditional alcohol and chlorine based substances are suggested as first choice during current COVID-19 supply crisis.
Very general guidelines are suggested for the disinfection of face shields [8] as following:
“Healthcare setting-specific procedures for cleaning and disinfecting used patient care equipment should be followed for reprocessing reusable eye protection devices. Manufacturers may be consulted for their guidance and experience in disinfecting their respective products. Contaminated eye protection devices should be reprocessed in an area where other soiled equipment is handled. Eye protection should be physically cleaned and disinfected with the designated hospital disinfectant, rinsed, and allowed to air dry. Gloves should be worn when cleaning and disinfecting these devices.”
Disinfection/cleaning recipes and methods that have been successfully and repeatedly tested for the designed shields in Fablab as following:
- Liquid A. Water solution of ethanol C2H5OH or isopropanol CH3CH(OH)CH3 96 wt.%.
- Liquid B. Water solution of chlorhexidine C22H30Cl2N10 0,3 wt. %.
The face shields spend 10-30 minutes fully immersed into the mixture of Liquid A and Liquid B having volume ratio A: B equal 2:1 to 2.2:1.8. To remove excessive liquid 100x bunch of face shields sped 4 minutes vertically to collect the liquid for next usage. Wiping with microfiber cloth allows to remove laser cutting contamination – it takes 8 second on each shield. At this speed some liquid remains on the surface and each next shield sticks to the stack. A complete set of 100 face shields and 100 head strips are sealed in an individual plastic bag with a heater. The residuals of disinfectant liquids are entrapped between PET parts providing prolonged sustained disinfection, what is especially suitable for urgent use in hospitals. The amount of liquid varies with relative wiping speed from 20ml to 28ml per 100 kits.
The design is also specifically suitable for fast sterilization when shields are re-used. Secondary disinfection is performed as following:
- Dispose contaminated elastic band thus “disassemble the face shield completely”, decontaminate 100% of the surface by rinsing to disinfection liquids and assemble it back with new elastic band (3 spare ones can be included into the package). It takes less than 5 minutes (30 seconds to disassemble, 2 for disinfection, 2 for re-assembling). If the face shield was used in “green” zone, the manual washing is recommended daily (2 minutes are enough alongside with hands washing).
- Order new the elastic bands in satisfactory amount per end-user facility (usually over 500 within one building) and receive additional packages.
Due to the use of reliable components, the service life of the shield is limited mainly by careless operation. In particular, it is important to carelessly store the " front surface on the table” - this creates scratches interfering comfortable view. The elastic band can be used for a long time without visible degradation. Careful operation should lead to a comfortable service life of about 1 working month.

Reviewer 3 Report
- The marking form of the figures should be consistent.
- The most important thing for medical supplies is to isolate bacteria, but no specific methods are proposed in this report.
Author Response
Reviewer #3.
We appreciate the valuable remarks from Reviewer and revised our manuscript in order to address them as following:
- The marking form of the figures should be consistent.
We checked all marks in all Figures and made them consistent in terms of used Font and parenthesis. We also revised the Caption to establish the consistent format.
- The most important thing for medical supplies is to isolate bacteria, but no specific methods are proposed in this report.
We agree with this remark and discuss this issue in the section REQUIREMENTS AND CONSTRAINTS adding the following phrases:
Disposable positive pressure isolation suit provides the highest (but not absolute) protection and it is actually used in “red” zones or in heavily infected spaces like residences for elderly persons where high morbidity rate was detected. This is an expensive and relatively scarce solution and hardly to be massively applied everywhere especially in the peak period.

Round 2
Reviewer 1 Report
Dear authors,
Many thanks for the very quick reply to my comments, the document has been enriched with more evidence and critical discussion. I would recommend you to embark upon a very meticulous proof-reading both in the text and references and maybe use-bullet points so as to make the content of the findings more presentable. Overall, this is very interesting and important research during these turbulent Covid-19 times and I would like to see the paper in print form.
I wish you all the best,